# Influence of Electrolyte Choice on Zinc Electrodeposition

**DOI:** 10.3390/ma17040851

**Published:** 2024-02-10

**Authors:** Kranthi Kumar Maniam, Corentin Penot, Shiladitya Paul

**Affiliations:** 1Materials Innovation Centre, School of Engineering, University of Leicester, Leicester LE1 7RH, UK; cp473@leicester.ac.uk; 2Materials Performance and Integrity Technology Group, TWI, Cambridge CB21 6AL, UK

**Keywords:** electrodeposition, zinc, ionic liquids, deep eutectic solvents, non-aqueous electrolytes, halide-free non-aqueous electrolytes, automotive, aerospace, marine, electroplating

## Abstract

Zinc electrodeposition serves as a crucial electrochemical process widely employed in various industries, particularly in automotive manufacturing, owing to its cost effectiveness compared to traditional methods. However, traditional zinc electrodeposition using aqueous solutions faces challenges related to toxicity and hydrogen gas generation. Non-aqueous electrolytes such as ionic liquids (ILs) and deep eutectic solvents (DESs) have gained attention, with choline-chloride-based DESs showing promise despite raising environmental concerns. In this study, zinc electrodeposition on mild steel was investigated using three distinct electrolytes: (i) halide-free aqueous solutions, (ii) chloride-based DES, and (iii) halide-free acetate-based organic solutions. The study examined the influence of deposition time on the growth of Zn on mild steel substrates from these electrolytes using physical characterization techniques, including scanning electron microscopy (SEM) and X-ray diffraction (XRD). The results indicate that glycol + acetate-based non-aqueous organic solutions provide an eco-friendly alternative, exhibiting comparable efficiency, enhanced crystalline growth, and promising corrosion resistance. This research contributes valuable insights into the impact of electrolyte choice on zinc electrodeposition, offering a pathway towards more sustainable and efficient processes. Through a comprehensive comparison and analysis of these methods, it advances our understanding of the practical applications of zinc electrodeposition technology.

## 1. Introduction

Zinc (Zn) electrodeposition is a key electrochemical process that holds immense significance across various industrial and technological domains, with a significant impact in the automotive sector [1,2]. The process of electrodeposition involves the formation of zinc and zinc-alloy coatings on metallic substrates, serving as a corrosion protection system and thereby extending the lifespan of critical mechanical components. Electrodeposition, often referred to as electroplating, represents an economical and efficient technology that plays a pivotal role in safeguarding and improving the functionality of metallic materials [3]. Unlike alternative methods such as hot-dipped galvanization, which results in thicker coatings, electroplating offers several advantages, including the ability to apply thinner coatings with a clean finish. These coatings not only bestow a smooth appearance but also confer corrosion resistance, all at a lower cost compared to many alternative coating methods [4].

Traditionally, Zn electrodeposition has been carried out using aqueous acid or alkali solutions, a process that, while effective, poses challenges due to the toxicity and corrosiveness of these electrolytes [5]. Additionally, these solutions are susceptible to concurrent hydrogen evolution reactions at the cathode that diminish the Faradaic efficiency of the electrochemical process. This limitation has spurred research into non-aqueous solutions, with a particular focus on room-temperature ionic liquids (ILs) and deep eutectic solvents (DESs). The use of ILs/DESs for Zn electrodeposition has gained considerable attention as a means to address the shortcomings associated with aqueous systems, such as the unwanted generation of hydrogen gas. These advanced coating solutions find applications in a diverse range of industries, including the marine and automotive sectors.

Research into Zn electrodeposition from choline chloride [6,7,8] and triflate-based DESs/ILs [9,10,11,12,13] has demonstrated a positive impact on the quality of the deposited coatings. Among the various ILs/DESs developed, choline-chloride-based systems have shown promise over conventional aqueous electrolytes [14,15,16,17]. However, these systems are not without their drawbacks, including the formation of chlorinated compounds and environmental toxicity concerns [18,19,20,21,22,23]. Consequently, the research focus has shifted towards the development of halide-free non-aqueous systems. Recently, acetate-based and propionate-based ILs/DESs, organic solutions have emerged as more viable alternative to halide-containing ones, offering a reduced environmental impact [24,25].

Non-aqueous solutions present a viable alternative to mitigate secondary reactions and achieve high-quality zinc coatings. Most of the previous work on the Zn electrodeposition from DESs reported in the literature focusses on choline-chloride-based combinations [15,26,27,28,29,30,31]. One of the key challenges in the use of most commonly used choline-chloride-based DESs lies in the high concentration of choline chloride, which often creates an aggressive environment towards both the substrate and the electrodeposit. Additionally, the hygroscopic nature of choline chloride necessitates a lengthy drying procedure before obtaining a DES. To overcome these issues, researchers have explored alternatives, such as modifying the choline chloride/ethylene glycol molar ratio in the bath or using ethylene glycol as a solvent with reduced chloride concentrations [32,33]. These approaches have shown promise, yielding high-purity films with improved corrosion performance.

However, ethylene glycol (EG)-based solutions come with their own set of challenges. They often exhibit lower metal solubility and lower solution conductivity. The use of ethylene glycol as a solvent with relatively low chloride concentrations has gained attention from multiple research groups [24,33,34,35,36]. This system offers a more straightforward experimental procedure, lower solution costs, and a less aggressive/corrosive environment towards the electrodeposited film and surrounding apparatus. However, as chloride concentrations decrease in the solution, the conductivity of the electrolyte diminishes. To maintain an inert environment with adequate conductivity, alternative precursor salts to chlorides with high solubility in EG must be explored. While the aforementioned works reflect on non-aqueous systems, studies on the comparative assessment of the non-aqueous system and the conventional aqueous system are scarce.

While acknowledging that EG may pose challenges such as increased viscosity and potential safety concerns related to toxicity, we believe that the overall benefits of using EG, as highlighted in the literature [36,37] and our own investigations, warrant consideration. The addition of additives to aqueous electrodeposition electrolytes to enhance zinc deposition properties, along with safety concerns arising from zinc precursor salts (e.g., zinc sulfate, zinc oxide, zinc chloride) and other electrolyte components (e.g., sodium/potassium chloride, sodium/potassium hydroxide, boric/acetic acid/ammonia), may increase the toxicity and viscosity of the electrolytes. Viscosity plays a critical role in dictating the coating nature and influencing the mass transportation of electroactive species, achieving a controllable deposit with good properties. Therefore, considering an electrolyte mixture of ethylene glycol and potassium acetate as environmentally responsible is reasonable for achieving Zn coatings.

The primary objective of this work is to investigate the electrodeposition of zinc onto mild steel substrates using three distinct electrolytes: (i) a halide-free aqueous (alkaline) solution, (ii) a chloride-based DES, and (iii) an acetate-based organic solution. This research aims to assess and compare the corrosion resistance, nucleation, and growth of the Zn coatings obtained from these different electrolytes, ultimately advancing our understanding of the most effective and environmentally friendly zinc electrodeposition methods.

This approach benchmarks the novel non-aqueous plating solutions against the halide-free conventional aqueous electrolytes. Acetate salts offer several advantages, including high solubility in ethylene glycol, providing an inert environment while maintaining sufficient conductivity for efficient electrodeposition processes. This study seeks to shed light on the efficacy of this approach in comparison to two established electrolyte systems: halide-free aqueous (alkaline) and chloride-based DES. By rigorously investigating and evaluating the corrosion performance, nucleation, and growth of Zn coatings obtained from these three distinct electrolytes, we aim to contribute valuable insights to the field of zinc electrodeposition. Ultimately, our research endeavors to facilitate the development of more environmentally friendly and efficient processes for enhancing the corrosion resistance of metallic materials, particularly in critical industries such as automotive manufacturing.

In the subsequent sections of this comprehensive study, we will delve into the experimental methods, results, and discussions, providing a detailed analysis of the electrodeposition processes and their respective outcomes. Through systematic comparisons and examinations, we aim to offer a deeper understanding of the advantages and limitations of each electrolyte, thereby advancing the knowledge base and practical applications of zinc electrodeposition technology.

## 2. Materials and Methods

Three distinct electrolyte solutions were prepared in-house for the electrodeposition of Zn, each with a unique composition and purpose: (i) halide-free alkaline aqueous (conventional), (ii) choline-chloride-based DES, and (iii) halide-free acetate-based organic solution.

### 2.1. Halide-Free Acetate-Based Organic Solution

The halide-free acetate solution was prepared by combining anhydrous ethylene glycol (EG, C_2_H_6_O_2_, Alfa Aesar (Louborough, Leicestershire, UK), 99.8%) and potassium acetate (CH_3_COOK, Fisher Scientific (Louborough, Leicestershire, UK )). The procedure followed was based on the method reported by Panzeri et al. [24], with modifications involving the use of potassium acetate as the conducting salt. First, EG was heated to 60 °C; then, a 0.5 M solution of CH_3_COOK was slowly added while maintaining the solution temperature at 60 °C. Next, zinc acetate (Zn (CH_3_COO)_2_·2H_2_O, Alfa Aesar, Louborough, Leicestershire, UK) with a concentration of 0.75 M was introduced into the solution, and thorough mixing was performed to achieve a homogeneous mixture.

### 2.2. Choline-Chloride-Based DES

The choline-chloride-based DES was prepared by combining choline chloride (ChCl, Sigma Aldrich (Gillingham, Dorset, UK)) and EG in a 1:2 molar ratio. The preparation process followed the procedure reported by Abbott et al. [38,39]. A 0.75 M solution of zinc chloride (ZnCl_2_, Alfa Aesar, Louborough, Leicestershire, UK, AR, >98%) was subsequently added to the mixture, and thorough mixing was carried out to ensure homogeneity.

### 2.3. Conventional Halide-Free Alkaline Aqueous Solution

The conventional halide-free alkaline aqueous solution was prepared by mixing 12 g/L of zinc oxide (ZnO, Sigma Aldrich) and 120 g/L of sodium hydroxide (NaOH, Alfa Aesar). This mixing process was carried out at 50 °C to ensure the formation of a homogeneous mixture. Photographs of the prepared solutions are shown in Appendix A.

### 2.4. Electrodeposition Process

Electrodeposition experiments were conducted using a Tenori Hull cell (Yamamoto-MS, Tokyo, Japan) with a test volume of 33 mL. A cell current of 0.1 A was applied for 10 min at 60 °C for the DES and organic solution experiments, whereas the conventional alkaline aqueous solution was tested at ambient conditions. To investigate the nucleation and growth of Zn coatings with time, a constant current of 10 mA cm^−2^ (1 A dm^−2^) was applied for 15 min and 30 min. In the electrodeposition process, mild steel was employed as the cathode for the Hull cell tests and constant current deposition, with zinc serving as the anode.

For the conventional alkaline zinc electrolyte, mild steel acted as the anode. Zn electrodeposition using the organic and non-aqueous media was carried out at 60 °C, whereas the deposition using the alkaline Zn solution was performed at room temperature. Prior to deposition, the mild steel cathode was subjected to etching in a 10% hydrochloric acid aqueous solution to ensure proper adhesion.

### 2.5. Post-Deposition Treatment

Following electrodeposition, the samples were cleaned by washing with DI water and acetone to remove any residual substances. Subsequently, the samples were dried using a compressed air gun and placed in a vacuum desiccator to eliminate moisture.

### 2.6. Cathode Current Efficiency (CCE)

The cathodic current efficiency (CCE) of the electrolytes with respect to Zn metal deposition was determined by gravimetric analysis at applied current densities of 10 mA cm^−2^ (1 A dm^−2^) for 15 min and 30 min. The CCE for Zn electrodeposition was calculated by measuring the change in weight (∆w) of the deposit on the mild steel cathode.

The CCE was calculated using the following equations:CCE, % = (∆w × 100)/w_t_(1)
w_t_ = I × t × M/(n × F)(2)
where ∆w is the change in weight after plating (g), w_t_ is the theoretical weight of the deposit (g), I is the current (A) passed in time (t in seconds), n represents the number of electrons transferred per Zn atom, M is the molecular weight of Zn (65.37 g/mol), and F is Faraday’s constant (96,485 C/mol) [40].

### 2.7. Physical and Electrochemical Characterization

The morphology of the deposited coatings over time was examined using optical microscopy and scanning electron microscopy (Zeiss ∑igma SEM (Cambridge, UK)) with energy-dispersive X-ray spectroscopy (Oxford Instruments X-Max^2^ detector, UK)). Microstructural investigations and particle size determinations were performed using X-ray diffraction (XRD). XRD was performed using an X-ray diffractometer (Model: Bruker AXS D8 Advance Karlsruhe, Germany); Source: Cu-kα) with a diffraction angle 2θ ranging from 10 to 90 degrees. Acquisition was made with a glancing angle incidence using Cu-Kα radiation (λ = 0.154 nm). The various phases were identified via the ICDD-JCPDS (International Center for Diffraction Data-Joint Committee on Powder Diffraction Standards) database.

Potentiodynamic polarization tests were conducted in a 3.5 wt% NaCl aqueous solution under natural aeration at ambient laboratory conditions (20–25 °C). The potential was swept from −0.2 V vs. open-circuit potential (OCP) at a scan rate of 10 mV min^−1^ to +0.2 V vs. OCP. The experimental setup consisted of a conventional three electrode cell, with platinum mesh as the counter electrode and a silver/silver chloride (Ag/AgCl, 3.5 M KCl solution) as the reference electrode. The samples were maintained at OCP for 30 min before starting the test.

## 3. Results and Discussion

The comparative analysis of the three electrolytes used in this study revealed intriguing insights into their performance during Zn electrodeposition. Samples after Hull cell deposition and depositions after 15 min and 30 min are visually represented in Appendix A. The representative optical micrographs are shown in Appendix A. Whereas slight burning was observed at high current density areas closer to the anode, bright deposits were observed throughout the Tenori Hull cell for the Zn deposited when employing three different electrolytes. For depositions at an applied current density of 10 mA cm^−2^ for 15 min or 30 min, the deposits were bright and had similar visual appearances. Uniform, smooth, and homogeneous coatings were observed in all the cases. Figure 1 compares the CCE of the three electrolytes at different times. Notably, the halide-free conventional alkaline aqueous solution exhibited the highest CCE among the three electrolytes, emphasizing its effectiveness as an electrodeposition medium when zinc is deposited for 15 min. It is worth noting that with an increase in time, the halide-free acetate solution could achieve CCE levels surpassing the chloride-based IL, indicating its favorable functionality in Zn electrodeposition. This finding reveals that the acetate solution’s capability to efficiently deposit Zn while being halide free, which is significant for reducing the environmental impact. This observation highlights the potential of acetate-based organic solutions as a promising alternative for Zn deposition, especially in environments where chloride-containing solutions may be undesirable due to their environmental impact or other constraints.

Figure 2 compares the SEM images of the deposited Zn utilizing the three different electrolytes at two different deposition times. As can be seen, the SEM images provided significant insights into the nature of the Zn deposits formed during this study. The results revealed that these deposits exhibited a compact and homogeneous structure, displaying distinct hexagonal features characteristic of Zn crystals. This observation suggests that the crystals exhibited enhanced growth over time, which is a crucial factor in determining the quality and efficiency of electrodeposition processes.

The comparative analysis of the three electrolytes used in this study revealed some interesting findings regarding their electrochemical performance. Initially, the halide-free conventional alkaline aqueous electrolyte exhibited the highest CCE among the three options. The ranking order of the CCE was as follows: halide-free aqueous > halide-free acetate > chloride-based DES.

However, a noteworthy shift in performance was observed with longer deposition times. The acetate-based organic solution started to demonstrate a relatively high percentage of CCE, which eventually approached the CCE value of the halide-free conventional alkaline aqueous electrolyte. This suggests that the acetate-based organic solution displayed remarkable functionality and adaptability over extended deposition durations, showcasing its potential as a viable alternative to conventional aqueous electrolytes.

In general, Zn in its metallic form adopts a conventional hexagonal close-packed structure. The morphology of the deposited zinc is guided by the favored crystal planes, which include [0 0 2], [1 0 0], and [1 0 1]. In particular, when the preferred orientation is [0 0 2], the resulting Zn sheet tends to grow at a slight angle, roughly between 0° and 30° with respect to the substrate [39]. This growth pattern leads to a deposition morphology that aligns in parallel with the substrate’s surface. The Zn deposition experiments performed using the three different electrolytes with varying deposition times were analyzed through XRD. The diffractograms depicted in Figure 3 exhibited prominent Zn [1 0 0], [0 0 2], and [1 0 1] peaks, with an increase in Zn [0 0 2] peak intensity observed as the deposition time extended. This indicates the evolution of the crystal structure during the deposition process.

It can be seen that the variation in the deposition time had two aspects of influence on the Zn coating when deposited using chloride-based DES. On one hand, with the increase in time for a given applied current density, the size of the Zn particles decreased. On the other hand, the increase in time reduced the surface flatness of the deposited zinc. Figure 3b shows that with the increase in the time, the deposited zinc will change from a state dominated by [1 0 1] crystal planes to a state with similar proportions of [1 0 1] and [1 0 0] planes. The crisscross of crystal planes with different orientations leads to the irregularity of the zinc film. The XRD pattern of the zinc coating changed greatly with the change in deposition time. In Figure 3, the strongest peak of the as-deposited zinc was the diffraction peak of the [1 0 1] crystal plane for a deposition time of 15 min using chloride-based DES, whereas for 30 min of deposition, the strongest peak corresponded to similar intensities of [1 0 0] and [1 0 1]. When the [1 0 0] and [1 0 1] crystal planes are dominant, the as-deposited zinc tends to grow at a large angle (about 70°~90°) with the substrate [41]. Therefore, the zinc deposited during short deposition times tends to grow parallel in the case of Zn deposition from conventional electrolytes, whereas it was perpendicular to the substrate in the case of chloride DES, since the dominant crystal plane is [1 0 1] in the latter. On increasing deposition time, in contrast, the deposited zinc tends to grow at large angles with the substrate because of the dominant crystal planes of [1 0 1] and [1 0 0] [42].

In the case of the acetate-based organic electrolyte solution, the XRD patterns exhibit a mix of [1 0 1] and [0 0 2], with an increasing intensity of [0 0 2] with an increase in the deposition time. Notably, the analysis of the full width at half maximum (FWHM) yielded distinct average crystallite sizes (t), as highlighted in Table 1. Zinc deposition using conventional halide-free alkaline aqueous electrolyte and the halide-free approach resulted in larger crystalline dimensions, indicative of crystalline growth over time. In contrast, deposition from chloride-based DESs led to a different crystalline dimension due to the lower peak intensity of [0 0 2] peaks. To quantitatively evaluate these differences, the Debye–Scherrer equation was applied. It revealed that the zinc deposits obtained from the conventional aqueous electrolyte had a crystalline dimension (d) of 42.9 nm, whereas the halide-free organic electrolyte approach resulted in a slightly larger crystalline dimension (d = 44.2 nm) after a 30 min deposition. Moreover, the purity of the deposited films was analyzed through energy dispersive spectroscopy (EDS), as presented in Appendix A. No oxygen or carbon impurities were detected, confirming the high-quality nature of the obtained films. These findings highlight the influence of the deposition conditions and electrolyte choice on the structural and purity characteristics of deposited zinc.

Furthermore, we have calculated the ratios of [0 0 2]/[1 0 0] and [0 0 2]/[1 0 1] from the intensities of the XRD peaks for the respective planes, and these values are presented in Table 1. This table provides XRD analysis data for the zinc coatings obtained with different electrolytes (considered in the study) for deposition times of 15 or 30 min.

The observed change in orientation of the zinc morphology, as indicated in the scanning electron micrographs (particularly in Figure 2b), is further corroborated by the X-ray diffractograms, which show a shift from the [0 0 2] plane to [1 0 0] and [1 0 1] planes. This change in crystal plane orientation could be attributed to the adsorption of complex ions (originating from chloride-based deep eutectic solvents (DES) and ZnCl_2_) during the crystal growth process.

The presence of complex ions, formed by the complexing reaction between chloride ions of varying concentrations present in choline chloride (ChCl) and from the ZnCl_2_ precursor, contributes to this effect. Previous reports on choline-chloride-based DES have confirmed that the presence of chloride ions alters the preferential orientation from the [0 0 2] plane to the [1 0 1], [0 0 2], and [1 0 2] planes, thereby influencing the morphological features [43], indicating varying degrees of the inhibition of the electrocrystallization process.

Therefore, it can be inferred that the presence of chloride ions in chloride-based DES and ZnCl_2_ likely influences the inhibition of the electrocrystallization process, resulting in changes in the morphology of zinc deposits and their orientation, as observed in Figure 2b,c and Figure 3a,b. This can be further witnessed from the [0 0 2]/ [1 0 0] and [0 0 2]/[1 0 1] ratios determined from the intensities of the respective XRDs and represented in Table 1, which resulted in the decrease in the crystallite size.

The corrosion behavior of zinc coatings deposited for 30 min using the three different electrolytes was evaluated through polarization tests conducted in 3.5 wt % NaCl solutions. The reason for using zinc coatings deposited for 30 min for the corrosion study was based on several factors. Firstly, we employed non-aqueous electrolytes for the deposition of zinc and observed an adequate zinc coating on the surface with good density, coverage, and mass after 30 min of deposition. This duration was deemed sufficient to achieve a stable and representative coating for corrosion testing.

Secondly, the objective of the study is to compare the corrosion performances of zinc coatings deposited using the aforementioned electrolytes for a given time. Therefore, it was essential to ensure consistency in the deposition time for all samples to facilitate meaningful comparisons between the different electrolytes.

Considering these factors, we opted to use the zinc coatings deposited for 30 min in the corrosion study to ensure uniformity and relevance to the study objectives. We believe that this approach allows for a reasonably good evaluation of the corrosion resistance of the zinc coatings obtained using various electrolytes involving aqueous, non-aqueous DESs and halide-free organic solution.

In a typical corrosion process, zinc dissolution involves the formation of an insoluble oxide/hydroxide film. However, in the presence of chlorides, this film tends to be porous and does not effectively inhibit zinc dissolution. Considering the surface texture, it was anticipated that the basal plane of the zinc coatings would exhibit better corrosion resistance compared to other surface orientations due to its more compact film morphology. The Tafel polarization plot in Figure 4 illustrates the corrosion behavior of the zinc deposits obtained from an acetate-based EG solution, resulting in an E_corr_ (corrosion potential) of approximately 1.02 V vs. Ag/AgCl and *j*_corr_ (corrosion current density) of 292 × 10^−6^ A.cm^−2^. These values were slightly lower than those observed for the choline-chloride-based DES. A summary of the representative corrosion values for coatings deposited using each electrolyte is presented in Table 2.

Notably, there are limited reports on the corrosion behavior of zinc electrodeposits from non-aqueous solutions [24,44,45], and some of those reports indicate similar or even worse performance compared to the findings in this study. Given the scarcity of comparative assessments between conventional aqueous electrolytes and choline-chloride-based DESs under similar conditions, this research provides significant understanding regarding the effectiveness of organic electrolytes when contrasted with traditional halide-free alkaline aqueous solutions. This sheds light on their corrosion tendencies and applicability across diverse use cases, offering valuable insights for practical implementation

## 4. Conclusions

In conclusion, this study provides an insight into the electrodeposition of zinc onto mild steel substrates using three distinct electrolytes: halide-free alkaline aqueous solutions, chloride-based DESs, and halide-free acetate-based organic solutions. The research aimed to compare their corrosion resistance, nucleation, and growth of zinc coatings, revealing significant impacts of electrolyte choice on the electrodeposition process. Initially, halide-free alkaline aqueous solutions showed the highest cathodic current efficiency (CCE); however, acetate-based organic solutions demonstrated adaptability with increasing deposition time from 15 min to 30 min, showcasing their potential as environmentally responsible alternatives. XRD analysis revealed differences in crystallite size and structure, emphasizing the influence of electrolyte choice. EDS confirmed the high purity of the deposited films, highlighting the importance of the deposition conditions and electrolyte choice. Polarization tests indicated promising corrosion resistance for zinc coatings deposited for 30 min from acetate-based organic solutions that was comparable to or better than chloride-based DES. The introduction of the Tenori Hull Cell, with its miniature volume, provides an industrial perspective, aiding visual quality assessment and reducing costs in dealing with non-aqueous electrolytes and offering a novel contribution to the field. The comparative analysis contributes to our understanding of electrolyte performance and supports advancements in the field. Overall, this study presents valuable insights into the impact of electrolyte choice on zinc electrodeposition, offering a pathway towards more sustainable and efficient processes and thus advancing the practical applications of zinc electrodeposition technology in various industries.

## Figures and Tables

**Figure 1 materials-17-00851-f001:**
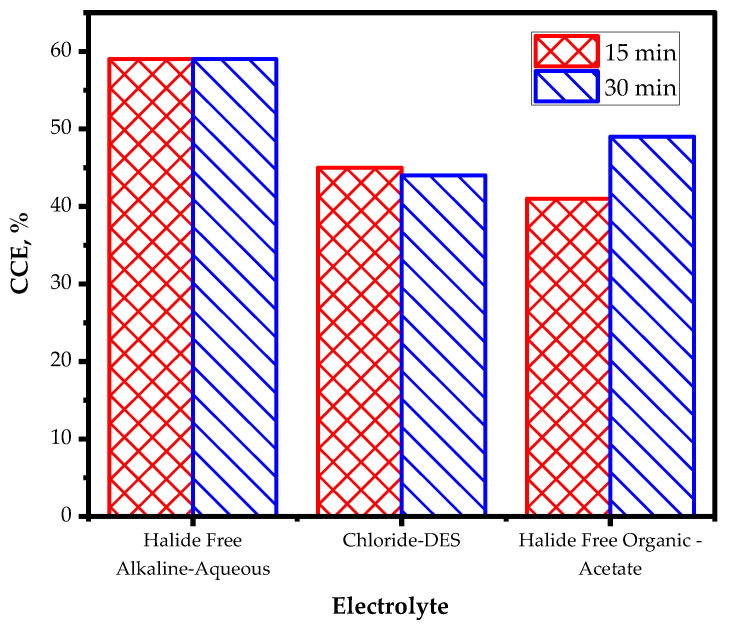
Figure comparing the cathodic current efficiency of the electrolytes after 15 min (red checked pattern) and 30 min (blue diagonal pattern) of Zn deposition.

**Figure 2 materials-17-00851-f002:**
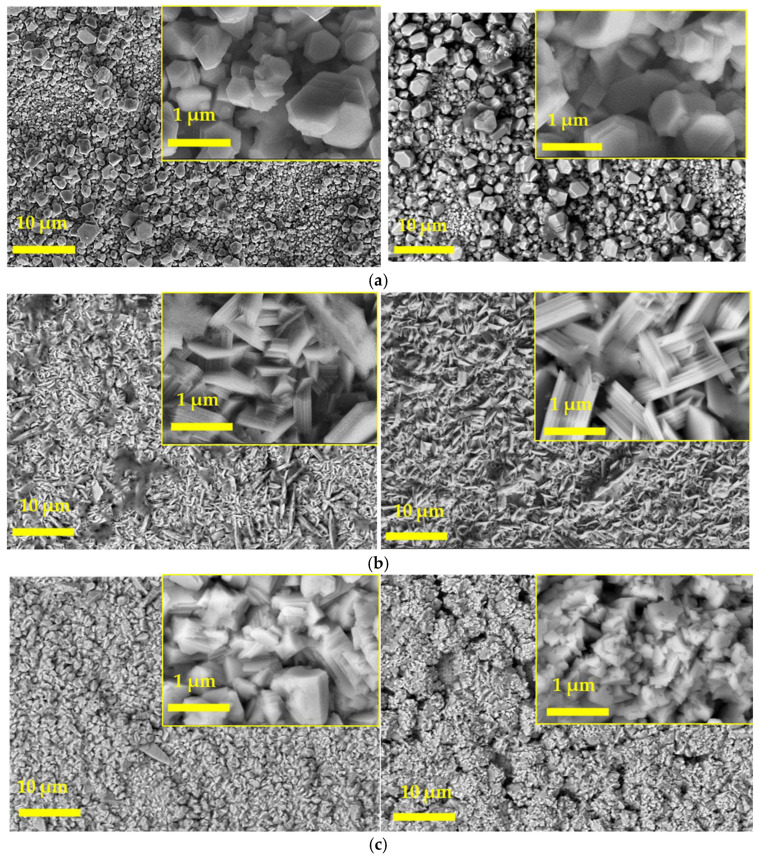
Figure comparing the scanning electron micrographs of the Zn deposited on mild steel substrates using (**a**) halide-free alkaline (conventional), (**b**) choline-chloride-based DES, and (**c**) acetate-based organic solution. (left)—zinc deposited for 15 min and (right)—zinc deposited for 30 min. Scale bar: 10 µm. Inset shows the higher magnification micrographs at a scale bar of 1 µm.

**Figure 3 materials-17-00851-f003:**
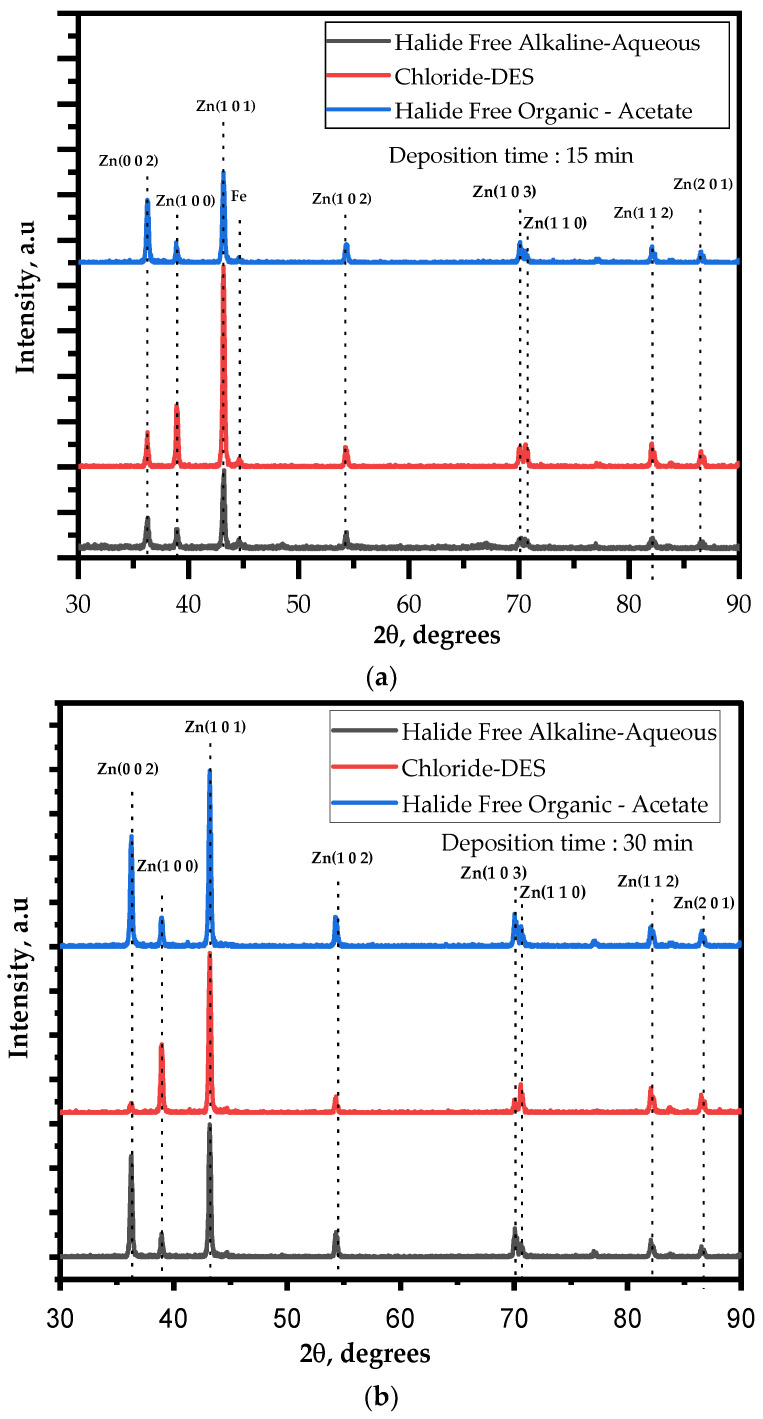
X-ray diffractograms of the Zn deposited on mild steel substrates for a deposition time of (**a**) 15 min and (**b**) 30 min using halide-free alkaline (conventional, black line); choline-chloride-based DES (red line) and acetate-based organic solution (blue line).

**Figure 4 materials-17-00851-f004:**
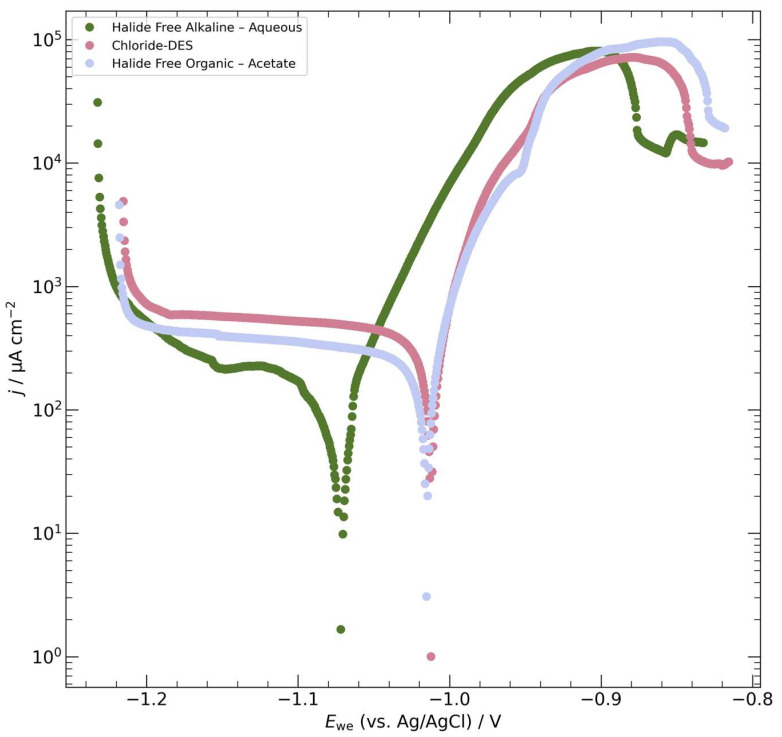
Comparative Tafel analysis of zinc coating performance in 3.5 wt% NaCl solution.

**Table 1 materials-17-00851-t001:** XRD analysis for zinc coatings obtained with different electrolytes for a deposition time of 30 min.

Electrolyte	Deposition Time, Minutes	FWHM (Degrees)	Average Crystal Size, d (nm)	[0 0 2]/[1 0 0]	[0 0 2]/[1 0 1]
Halide-Free Alkaline—Aqueous	15	0.2555	38.6	1.624	0.404
	30	0.2299	42.9	4.451	0.776
Chloride DES	15	0.2108	46.7	0.558	0.171
	30	0.3082	32	0.167	0.071
Halide-Free Organic—Acetate	15	0.2232	41.9	2.873	0.684
	30	0.2356	44.2	3.710	0.634

**Table 2 materials-17-00851-t002:** Tafel plot results for zinc coatings obtained with different electrolytes for a deposition time of 30 min.

Electrolyte	E_Corr_, V (vs. Ag/AgCl)	*j*_Corr_, µA cm^−2^
Halide-Free Alkaline—Aqueous	−1.07	139.48
Chloride DES	−1.01	295.88
Halide-Free Organic—Acetate	−1.02	292

## Data Availability

Data are contained within the article and Appendix A.

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
