# Peer review of "Influence of Electrolyte Choice on Zinc Electrodeposition"

_materials, 2024, doi:10.3390/ma17040851_

Round 1

Reviewer 1 Report

Comments and Suggestions for Authors

This manuscript is a good attempt to apply a halide-free organic solution for Zn electrodeposition. The usage of the acetate-based salt and ethylene glycol solvent provided a better strategy for electrodeposition than conventional chloride-based aqueous solutions. Although the authors already provided sufficient data and analysis, they still need to address some issues and concerns. Therefore, it is suggested to accept this work after a minor revision. The comments and suggestions about this work are described as follows:

1. The authors claimed that the usage of KAc and EG is more environmentally friendly. However, compared with H2O, EG solvent may increase the viscosity and decrease the ionic conductivity. Meanwhile, the flammability and toxicity of EG may provide other safety concerns and toxic issues to the environment. The authors should discuss these concerns in the manuscript.

2. On page 3, the authors provided the name of potassium acetate but used the chemical formula of sodium acetate (CH3COONa). The authors should double-check and provide the correct chemical compounds they used in this work.

3. The scale bars in Figure 2 are too blurry, especially the inset figures in Figure 2a and 2b. It is suggested that the authors draw the scale bars themselves instead of directly using the scalebars from the SEM instrument. Moreover, in Figure 2b and 2c, the authors provided the morphologies under a scalebar of 10 μm, whereas we can only find the image under a scalebar of 2 μm in Figure 2a. It is highly suggested to provide an SEM image under 10 μm to help readers evaluate the morphology difference. 

4. In Figure 3, the authors provided the XRD patterns of Zn after deposition. It is interesting to see the preferential exposure of [002] orientation in the acetate-organic solvents. However, in Figure 2b, the overlap of each curve hides the information on peak intensities. Therefore, it is suggested to redraw the figures to show all the peaks clearly without any overlap. Further, it is suggested to provide the ratios of [002]/[100] and [002]/[101], which will provide more intuitive results for readers to evaluate the deposition.

5. The authors should check the acronyms in the manuscript. After defining the acronyms in the first place, they don’t need to write the full name again.

6. The authors are suggested to zoom-in all the scale bars in all the SEM images, especially in Figure S3.

7. It is suggested to re-write the abstract with more details. For instance, they said “this work contributes valuable insights into the mechanism of deposition process,” whereas they did not specify the mechanism here. They should add more details in the abstract to attract the attention of the readers.

8. I did not see very detailed insights into the mechanism in their optimized electrolyte. Why the addition of EG can reorientate the Zn deposition? Which is more important to regulate the Zn deposition? EG or acetate? The authors should discuss more about the mechanism if they want to claim they have insights into the electrodeposition mechanism.

Comments on the Quality of English Language

The authors should modify their English minorly.

Reviewer 2 Report

Comments and Suggestions for Authors

The paper investigates the electrodeposition process of zinc where three different electrolytes. The conclusion highlights the acetate based solutions as an eco-friendly alternative to actual used electrolytes. The electrolytes used in this paper are:

1.      (Zn (CH3COO)+CH3COONa – I guess here is an error. Na in the formula but potassium acetate is mentioned in the text.

2.      ChCl + ZnCl2

3.      ZnO + NaOH

 Th electrodeposition processes and the post treatments are first described. The Zn deposits are then compared (also in the Supplementary Material)

-        -   Morphology (SEM+XRD)

-         -  Electrochemical performances (CCE + taffel plots)

TThen some conclusions are drawn. The paper is well written, and conclusion are just (somes small inadvertences Ex: different results in crystallite size cannot be due to lower peak intensities. The lower XRD intensities is an effect not a cause),

However, what is completely missing in this paper is the novelty. As even the authors mention

 1st electrolyte “The procedure followed was based on the method reported by Panzeri et al.[24],”

2nd electrolyte “The preparation process followed the proce-dure reported by Abbott et al.[37,38].”

Concerning the 3rd electrolyte, this was first time reported at least 15 years ago (see J. Cent. South Univ. Technol. (2007)01−0037−05, DOI: 10.1007/s11771−007−0008−1, Mechanism of zinc electroplating in alkaline zincate solution, PENG Wen-jie WANG Yun-yan. Those authors performed a more complex study on the electroplating process.  

Reviewer 3 Report

Comments and Suggestions for Authors

The authors report on the Zn electrodeposition process with different types of electrolytes. The influence of deposition time with different electrolytes on the Zn deposition behavior was systematically studied and characterized with SEM, XRD, and electrochemical testing. I recommend acceptance of the manuscript after resolving the following comments.

1) Please provide more details in the Materials and Methods, e.g., the ratio between EG and CH3COONa, and SEM working conditions etc.

2) Consider plotting Figures 1a and b into 1 figure with different colors for a direct comparison of current efficiency.

3) In Figure 2, scalebars are missing in some of the enlarged figures and please add EDS mapping of the collected SEM images. In addition, can you explain further why the orientation of the Zn flask looks different, especially in Figure 2B?

4) Figure 3 should be plotted as a stacked image with intensity change information of each Zn plane upon longer reaction time. Please add the standard Zn PDF card as well.

5) Why does the Zn crystal size decrease with a longer coating time for Chloride-DES electrolyte?

6) Please explain why the zinc coatings deposited for 30 minutes were used for the corrosion study instead of the samples coated for 15 minutes.

Round 2

Reviewer 2 Report

Comments and Suggestions for Authors

Using a different Hull Cell is not exactly a reason for novelty. These cells are adapted for lower volumes (used specially for noble metals). However, overall effort is good.